# Assessing interface accuracy in macromolecular complexes

Olgierd Ludwiczak[1], Maciej Antczak[1,2]*, Marta Szachniuk [1,2]*

**1** Institute of Computing Science, Poznan University of Technology, Poznan, Poland, **2** Institute of Bioorganic Chemistry, Polish Academy of Sciences, Poznan, Poland

\* mantczak@cs.put.poznan.pl (MA); mszachniuk@cs.put.poznan.pl (MS)

## Abstract

Accurately predicting the 3D structures of macromolecular complexes is becoming increasingly important for understanding their cellular functions. At the same time, reliably assessing prediction quality remains a significant challenge in bioinformatics. To address this, various methods analyze and evaluate *in silico* models from multiple perspectives, accounting for both the reconstructed components' structures and their arrangement within the complex. In this work, we introduce Intermolecular Interaction Network Fidelity (I-INF), a normalized similarity measure that quantifies intermolecular interactions in multichain complexes. Adapted from a well-established score in the RNA field, I-INF provides a clear and intuitive way to evaluate the predicted 3D models against a reference structure, with a specific focus on interchain interaction sites. Additionally, we implement the $F_1$ measure to assess interfaces in macromolecular assemblies, further enriching the evaluation framework. Tested on 72 RNA-protein decoys, as well as exemplary DNA-DNA, RNA-RNA, and protein-protein complexes, these measures deliver reliable scores and enable straightforward ranking of predictions. The tool for computing I-INF and $F_1$ is publicly available on Zenodo, facilitating large-scale analysis and integration with other computational systems.

## Introduction

Molecular complexes play crucial roles in various cellular processes, including gene expression and homeostasis. Understanding their biological functions relies on detailed structural studies that reveal the conformations of constituent molecules and the mechanisms by which they form stable complexes. Traditionally, such studies have employed experimental techniques, however, in recent years, computational prediction methods have gained prominence, generating increasingly accurate and reliable models. These methods undergo systematic evaluation in blind prediction challenges, such as CASP or RNA-Puzzles, where computational predictions are assessed both in the context of reference structures and independently from stereochemistry and chain topology perspectives [1–5].

Evaluating molecular complex predictions is inherently challenging and often involves a variety of scoring functions, including knowledge- and machine-learning-based approaches [6]. Some of these functions were initially developed for isolated proteins or nucleic acids and

**Data availability statement:** The tool for computing I-INF and F1 to assess interfaces in macromolecular assemblies, along with a user manual and ready-to-run examples, is publicly

available at Zenodo (https://dx.doi.org/10.5281/zenodo.14697284) and GitHub (https://github.com/OlgierdL/iinf). For benchmarking, we used the dataset available at https://zoulab.dalton.missouri.edu/RNAdecoys/index.html.

**Funding:** The author(s) received no specific funding for this work.

**Competing interests:** The authors have declared that no competing interests exist.

later adapted to assess their complexes. For example, RMSD was adapted to score the interface between various chains of predicted multimeric assemblies and was applied as I-RMSD (Interface RMSD) to RNA-ligand predictions [7]. Along with LRMSD (Ligand RMSD), I-RMSD has become part of DockQv2 [8], which is aimed at evaluating the accuracy of complexes involving proteins, nucleic acids, and small molecules. Other scoring functions developed specifically for multimers include oligolDDT [9] and US-align [10]. In general, the evaluation of molecular complexes proceeds in two ways: by assessing the quality of interactions between chains or overall structural similarity.

Intermolecular Interaction Network Fidelity (I-INF), introduced here to assess the prediction of macromolecular complexes, is an adaptation of the RNA-specific INF score [11], with the focus shifting from base pairs to intermolecular interactions. Like the original, I-INF is a similarity measure ranging from 0 to 1, where 0 indicates a completely incorrect prediction and 1 signifies that the prediction is fully consistent with the reference structure. Tested on 72 structures from RNA-protein docking decoys [13], I-INF shows a high correlation with TM-score-based rankings [14] and a low correlation with DockQv2 [8]. This complementary nature highlights I-INF's usefulness as part of a broader toolkit for evaluating three-dimensional macromolecular models. To provide additional flexibility for users, we also implement the $F_1$ score, which is widely used for evaluating protein structure predictions, particularly focusing on hydrogen bonds [12]. While both I-INF and $F_1$ assess the same aspect of macromolecular assemblies, their mathematical formulations differ: I-INF uses a geometric mean, whereas $F_1$ employs a harmonic mean. Similar to I-INF, we adapt $F_1$ to evaluate interchain interactions; in both measures, an interaction is counted as a true positive regardless of the number of hydrogen bonds in the predicted model. Together, I-INF and $F_1$ provide a robust framework for assessing intermolecular interfaces in macromolecular complexes.

## Materials and methods

Data processing for computing I-INF and $F_1$ involves three steps: preprocessing the 3D structure data, quantifying hydrogen bonds that form intermolecular interactions in both the predicted and native structures, and rescaling the score based on the target coverage by predicted residues (Fig 1).

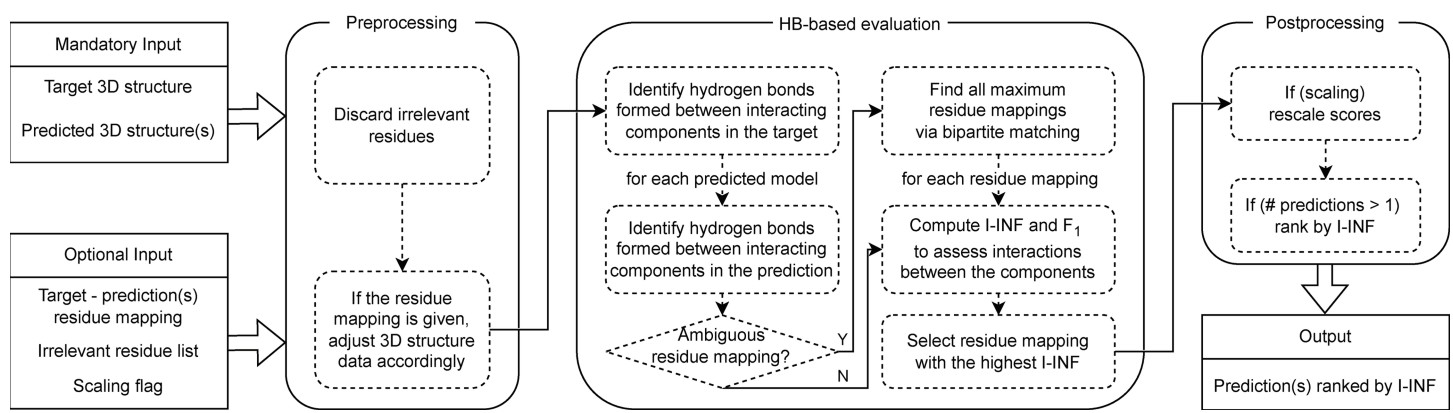

**Fig 1. Data flow in the assessment of macromolecular assembly predictions.**

In the input stage, users provide the reference and predicted 3D structure(s) in PDB format. They may also ensure consistent residue mapping between each model and the reference structure as well as provide additional information about irrelevant residues and a scaling flag. Preprocessing begins by filtering out irrelevant residues in the input structures, if specified. Next, the 3D structure data are processed using rna-tools [15] to align with the provided residue mapping, if available. In the third phase, HBPLUS [16] is executed to identify hydrogen bonds between RNA, DNA, and protein chains in each molecular assembly. If residue mapping between the predicted model and the target is not predefined, an additional algorithm is employed to determine all maximum mappings. This algorithm operates on a bipartite graph that represents the sequences of the analyzed structures and identifies the maximum bipartite matching within the graph [17]. Subsequently, pairs of binding residues are extracted from both the target and predicted models regardless of the number of hydrogen bonds they form. Each residue pair is categorized as a true positive (TP; present in both the target and prediction), false positive (FP; present only in the prediction), or false negative (FN; present only in the target). Finally, the I-INF and $F_1$ scores are computed for the predicted model:

$$I - INF = \sqrt{\frac{|TP|}{|TP| + |FP|} * \frac{|TP|}{|TP| + |FN|}}$$

$$F_1 = \frac{|TP|}{|TP| + \frac{1}{2} * (|FP| + |FN|)}$$

After calculating the scores, if multiple mappings exist for a predicted structure, only the one with the highest I-INF (and $F_1$) score is selected for the subsequent steps. Postprocessing involves the optional rescaling of the I-INF and $F_1$ values by multiplying them by the fraction of predicted residues contained in the target. For example, if a target has 100 residues and 80 of those are predicted in the model, the scale factor would be 0.8. The predicted models are then sorted in non-increasing order by I-INF, and a list of model names with their assigned I-INF and $F_1$ values is output in a CSV file.

The I-INF tool was developed in Python 3 and is available under the MIT license, with ready-to-run examples. It is published on GitHub and Zenodo, and supports the processing of various types of intermolecular complexes (e.g., RNA-RNA, DNA-DNA, protein-protein).

## Results and discussion

To test and validate I-INF, we applied it to evaluate RNA-protein docking decoys, consisting of 72 experimental 3D structures of varying complexity along with their *in silico* generated models. Fig 2 shows a sample model-target pair from this collection, highlighting the intermolecular interactions. In the target structure (PDB ID: 3MOJ [18]), there are 4 interactions forming hydrogen bonds, whereas, in the predicted model, there are 9, of which 4 are true positives and the remaining 5 are considered false positives. In this case, I-INF is 0.67.

For comparison, we evaluated all models from the benchmark set using two other methods, TM-score [14] and DockQv2 [8]. Both are normalized similarity measures that take values between 0 and 1. The numerical results of this comparative analysis are provided in S1 Table. Fig 3 visualizes the correlations between TM-score, DockQv2, and I-INF, while S1 Fig illustrates the distribution of these measures across the benchmark set. We also analyzed the

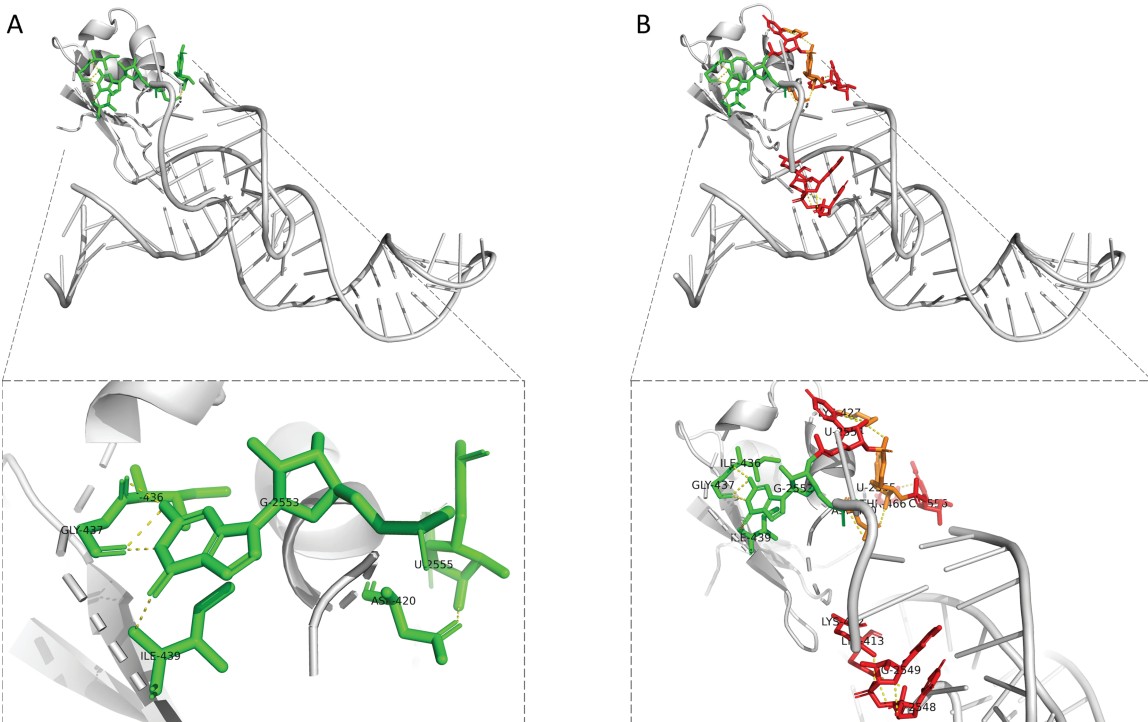

**Fig 2. (A) Reference structure (PDB ID: 3MOJ) and (B) the predicted model of the RNA binding domain of the Bacillus subtilis YxiN protein complexed with a fragment of 23S ribosomal RNA.** Residues involved in RNA-protein binding are color-coded: green for true positives (interactions present in both the reference structure and the model), red for false positives (interactions present only in the predicted model), and orange for 3 residues that form a multiplet in the predicted model, where one interaction is a true positive and the other is a false positive. No false negatives (interactions present only in the reference structure) are observed for this pair of structures.

Pearson correlation between the score rankings generated by these three measures. Although TM-score evaluates the global topology of the model and I-INF specifically assesses the accuracy of the intermolecular interface, we observed a high Pearson correlation (0.73) between these metrics. This suggests that, in our dataset, a correctly predicted global fold is largely a consequence of the proper spatial arrangement of the molecular components, resulting in accurately modeled interfaces. In other words, deviations in the overall fold are primarily associated with errors in the interface regions. Therefore, high TM-score values frequently coincide with high I-INF values, highlighting our models' interdependence between global structural accuracy and interface correctness. In contrast, the low correlation with DockQv2 (0.18) indicates that it provides different insights than I-INF. Thus, these two measures are complementary, and it is beneficial to use both in evaluations.

In an additional computational experiment, we analyzed the results of two RNA-protein targets from CASP15. The native assemblies consisted of one RNA strand and six protein chains for RT1189 (PDB ID: 7YR7 [19]) and one RNA strand and four protein chains for RT1190 (PDB ID: 7YR6 [19]). While the predicted complexes often included accurate protein or RNA structures, their chains were not properly docked, as indicated by low TM-score values (none of the models exceeded 0.5). This was further confirmed by I-INF calculations, which were close to zero in all cases.

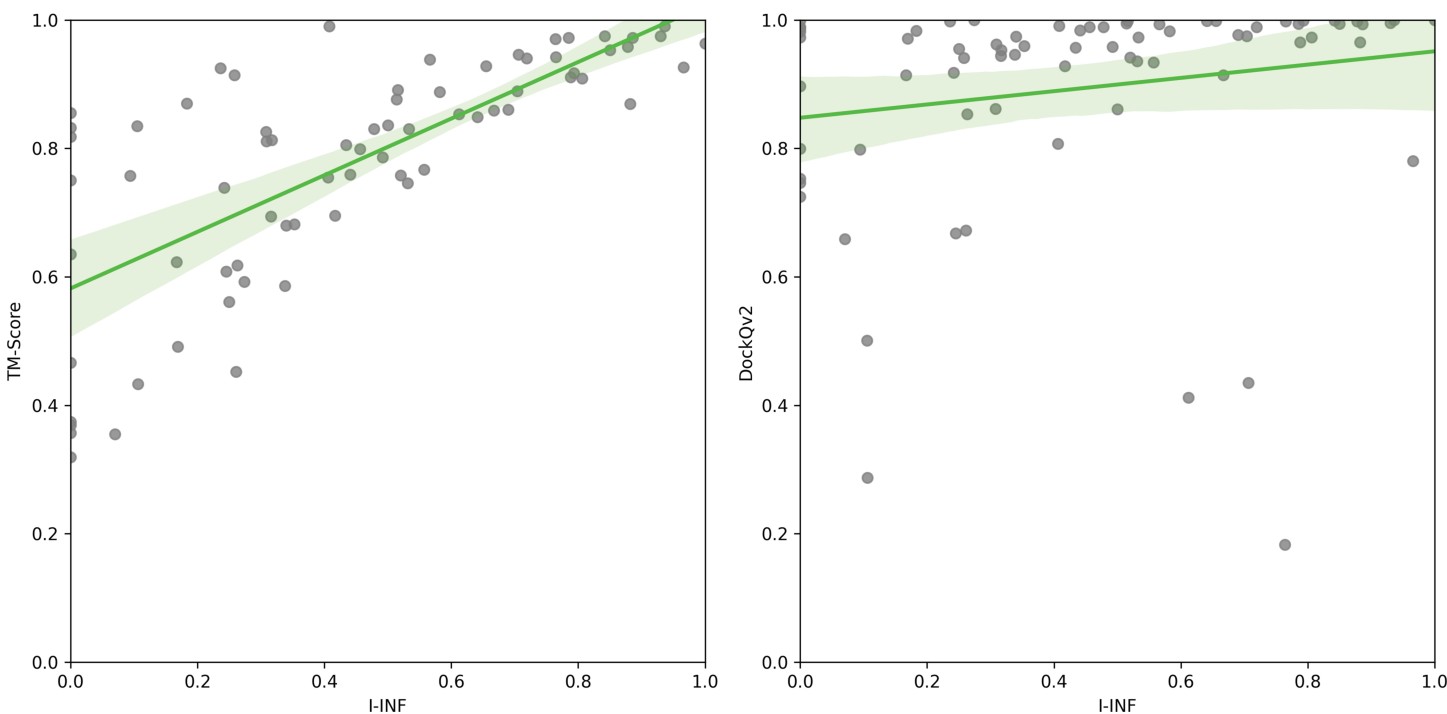

**Fig 3. Correlation between TM-score and I-INF (left) and DockQv2 and I-INF (right).**

## Conclusions

In this study, we presented the I-INF metric (Intermolecular Interaction Network Fidelity) as a novel approach for evaluating the accuracy of intermolecular interactions within macromolecular complexes. I-INF provides a complementary measure to existing scoring functions, such as TM-score and DockQv2, offering a robust tool for assessing the interfaces in RNA-protein and other macromolecular complexes. Additionally, we have incorporated the $F_1$ as part of our evaluation process, allowing for a more comprehensive comparison with other commonly used measures in structural prediction.

The tool for calculating I-INF and $F_1$ operates on input PDB files, which is currently the only supported format due to limitations of HBPLUS. In future updates, we plan to expand its functionality by adding support for the mmCIF format, which will enhance compatibility with other widely used structural databases and prediction tools. As the quality of predictions improves, we may increase the sensitivity of our approach by focusing on hydrogen bonding interactions, rather than only on residue pairs involved in these interactions.

## Supporting information

**S1 Table. Evaluation of the predicted 3D models from the RNA-protein docking decoys.**
(PDF)

**S1 Fig. A distribution of TM-score, DockQv2, and I-INF values computed for the benchmark set.**
(PDF)

## Acknowledgments

This work was carried out at Poznan University of Technology (https://www.put.poznan.pl/en) and the Institute of Bioorganic Chemistry, Polish Academy of Sciences (https://www.ibch.poznan.pl/en.html). The authors are grateful for the support and resources provided by the institution.

## Author contributions

**Conceptualization:** Maciej Antczak, Marta Szachniuk.

**Data curation:** Maciej Antczak.

**Formal analysis:** Marta Szachniuk.

**Methodology:** Maciej Antczak, Marta Szachniuk.

**Resources:** Marta Szachniuk.

**Software:** Olgierd Ludwiczak, Maciej Antczak.

**Supervision:** Marta Szachniuk.

**Validation:** Maciej Antczak, Marta Szachniuk.

**Visualization:** Maciej Antczak, Marta Szachniuk.

**Writing – original draft:** Marta Szachniuk.

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
