## [Decision Letter · Decision Letter 0]

15 Dec 2024

PONE-D-24-53421The I-INF metric for assessing the quality of macromolecular assembly predictionPLOS ONE

Dear Dr. Szachniuk,

Thank you for submitting your manuscript to PLOS ONE. After careful consideration, we feel that it has merit but does not fully meet PLOS ONE’s publication criteria as it currently stands. Therefore, we invite you to submit a revised version of the manuscript that addresses the points raised during the review process.

The reviews highlight both the potential utility of I-INF and several significant concerns regarding its novelty, theoretical foundation, and applicability. While one reviewer expressed overall satisfaction, their sole recommendation—a comparative figure—should be considered alongside the more extensive issues raised by the other reviewer. The manuscript provides a valuable contribution to the evaluation of structural prediction methods. However, addressing the outlined deficiencies is essential to ensure the robustness, usability, and contextual grounding of I-INF as a broadly applicable metric.

We look forward to receiving your revised manuscript.

Kind regards,

Yong Wang

Academic Editor

PLOS ONE

Journal Requirements:

3. Thank you for uploading your study's underlying data set. Unfortunately, the repository you have noted in your Data Availability statement does not qualify as an acceptable data repository according to PLOS's standards. At this time, please upload the minimal data set necessary to replicate your study's findings to a stable, public repository (such as figshare or Dryad) and provide us with the relevant URLs, DOIs, or accession numbers that may be used to access these data. For a list of recommended repositories and additional information on PLOS standards for data deposition, please see https://journals.plos.org/plosone/s/recommended-repositories.

Reviewers' comments:

Reviewer's Responses to Questions

**Comments to the Author**

1. Is the manuscript technically sound, and do the data support the conclusions?

Reviewer #1: Partly

Reviewer #2: Yes

2. Has the statistical analysis been performed appropriately and rigorously? 

Reviewer #1: Yes

Reviewer #2: Yes

3. Have the authors made all data underlying the findings in their manuscript fully available?

Reviewer #1: Yes

Reviewer #2: Yes

4. Is the manuscript presented in an intelligible fashion and written in standard English?

Reviewer #1: Yes

Reviewer #2: Yes

5. Review Comments to the Author

Reviewer #1: This manuscript describes I-INF, a metric to evaluate the correctness of inter-molecular hydrogen bonding. Both the code and the manuscript is overly simple. Major issues include:

1. Hydrogen bonding evaluation is routinely done in CASP, e.g., in https://pmc.ncbi.nlm.nih.gov/articles/PMC4282348/, except that I-INF takes the geometric average while CASP uses harmonic average. Please properly credit earlier works and explain the differences.

2. TM-score an I-INF does NOT measure the same aspect of complex assembly. I-INF only evaluate the correctness of interface, while TM-score evaluates the global topology. There should be some theoretical justification on why TM-score has a high correlation with I-INF.

3. The title strongly suggest that I-INF works even for any bio-macromolecular complexes including pure protein complexes. However, all evaluations in the result section are on protein-RNA complexes. There should be some application of I-INF on protein-protein complexes or explanation on why this is not applicable.

4. A major challenge in complex assembly evaluation is symmetric. For example, suppose the input is a tetramer containing two identical copies of a protein and two identical copies of an RNA, where the two RNA chains interact with each other. In this case, deriving a correct chain mapping for optimal I-INF is not trivial. This work simply ignore this challenge by forcing the user to input consistent chain ID mapping. This makes the program less useful.

5. There is no support for mmCIF format, which is particularly relevant for this work which is supposed to work with large complexes with >65 chains.

Reviewer #2: This work studies the evaluation of 3D structural prediction for macromolecular complexes, which is an essential problem in understanding cellular functions. The authors propose a novel metric, I-INF (Intermolecular Interaction Network Fidelity), to quantitatively assess the intermolecular interactions within multichain complexes. This manuscript is very well written and the I-INF score is useful. And I have only one minor comment about it.

When comparing I-INF with existing methods such as TM-score and DockQv2, it is recommended to add a figure to make the comparison more clear.

6. PLOS authors have the option to publish the peer review history of their article (what does this mean?). If published, this will include your full peer review and any attached files.

Reviewer #1: No

Reviewer #2: No

---

## [Author Response · Author response to Decision Letter 1]

29 Jan 2025

We thank the Reviewers for all the comments and suggestions. We responded to them to the best of our ability and made the required changes to the manuscript.

In line with PLOS ONE recommendations, we integrated the Zenodo platform with our tool repository on GitHub, enabling us to assign a unique DOI for each new release. The current DOI for the I-INF tool (version 1.0) is 10.5281/zenodo.14697284, and it can be accessed via https://zenodo.org/records/14697284 or https://dx.doi.org/10.5281/zenodo.14697284. All future updates will also be made available through these links.

Reviewer 1

This manuscript describes I-INF, a metric to evaluate the correctness of inter-molecular hydrogen bonding. Both the code and the manuscript is overly simple. Major issues include:

{Comment 1:} Hydrogen bonding evaluation is routinely done in CASP, e.g., in https://pmc.ncbi.nlm.nih.gov/articles/PMC4282348/, except that I-INF takes the geometric average while CASP uses harmonic average. Please properly credit earlier works and explain the differences.

{Response:} We thank the Reviewer for highlighting the evaluation approach used in CASP, particularly the application of the harmonic mean (e.g., F1 score) for assessing hydrogen bond predictions. While CASP assessors commonly employ the harmonic mean (F1 score), the RNA-Puzzles contest has predominantly utilized the geometric mean (INF score) for evaluating RNA-related predictions. A key distinction between the evaluation methods in CASP and RNA-Puzzles lies in the granularity of interaction assessment. In CASP, individual hydrogen bonds are explicitly evaluated when calculating the F1 score. In contrast, RNA-Puzzles focuses on broader interaction assessments rather than individual hydrogen bonds. For instance, if two nucleotides in the reference structure are connected by three hydrogen bonds, but the predicted model shows the same nucleotides forming only two hydrogen bonds, the interaction is still considered a true positive. This approach disregards discrepancies in the exact number of hydrogen bonds, focusing instead on the presence of the interaction itself rather than its precise details. Such a solution seems more suitable for predictions that have not yet reached exceptional accuracy.

In response to the Reviewer’s comment, we have implemented the F1 score in our scripts to enable direct comparison with I-INF. In both I-INF and F1 measures, an interaction is counted as a true positive regardless of the number of hydrogen bonds in the predicted model. Consequently, we updated Figure 1 (depicting the workflow) and revised the description of our program for molecular complex evaluation to reflect this addition. We also updated the manuscript title to emphasize the inclusion of multiple evaluation methods and our commitment to presenting a balanced and comprehensive analysis.

Finally, it is worth noting that when evaluating multiple predictions, our program ranks them based on the value of the evaluation function. We designated I-INF as the primary arbiter, as substituting it with F1 would not affect the final ranking.

{Comment 2:} TM-score an I-INF does NOT measure the same aspect of complex assembly. I-INF only evaluate the correctness of interface, while TM-score evaluates the global topology. There should be some theoretical justification on why TM-score has a high correlation with I-INF.

{Response:} We appreciate the comment of the Reviewer on the correlation between TM-score and I-INF. In our opinion, the justification for the observed high correlation between these two measures is closely tied to the characteristics of the validation dataset. In general, the tertiary structures of the models closely resemble the reference structures, with the primary differences occurring at the interface regions, which is the focus of our evaluation.

The validation set was specifically designed to assess the quality of methods evaluating intermolecular interfaces. Since I-INF is focused on assessing the accurracy of the interfaces, while TM-score evaluates the global topology, the significant overlap in terms of structural correctness at the interface region likely accounts for the high correlation between the two measures. This makes sense, as both metrics are sensitive to the accuracy of structural predictions, albeit with different scopes of evaluation.

{Comment 3:} The title strongly suggest that I-INF works even for any biomacromolecular complexes including pure protein complexes. However, all evaluations in the result section are on protein-RNA complexes. There should be some application of I-INF on protein-protein complexes or explanation on why this is not applicable.

{Response:} We thank the Reviewer for this observation. First, as mentioned in our response to Comment 1, we have updated the manuscript title to remove the explicit reference to I-INF. This change ensures that the title no longer implies an exclusive focus on this measure.

Second, we agree that I-INF is not inherently limited to RNA-protein complexes. The measure is designed to evaluate the interface accuracy in any type of macromolecular complex. To demonstrate this, we have now included additional examples that apply I-INF to DNA-DNA, RNA-RNA, and protein-protein assemblies. They are available in our Zenodo repository (DOI: 10.5281/zenodo.14697284). This addition underscores the versatility of the I-INF measure and further substantiates its broader applicability beyond RNA-related systems.

{Comment 4:} A major challenge in complex assembly evaluation is symmetric. For example, suppose the input is a tetramer containing two identical copies of a protein and two identical copies of an RNA, where the two RNA chains interact with each other. In this case, deriving a correct chain mapping for optimal I-INF is not trivial. This work simply ignore this challenge by forcing the user to input consistent chain ID mapping. This makes the program less useful.

{Response:} We thank the Reviewer for raising this important point regarding the challenge of symmetric complexes. We acknowledge that deriving a correct chain mapping for complex assemblies, especially in cases of identical components such as in the example of tetramers, can be non-trivial. To address this issue, we have implemented a mechanism similar to that used in DockQv2. Specifically, we introduced an algorithm that efficiently iterates over all possible combinations of chains that can occur between the model and the reference structure. For each model, we now select the highest I-INF value from all analyzed chain mappings.

To achieve this, we construct a bipartite graph that includes two independent layers of vertices corresponding to chains in the model and the reference structure. An edge is created between two vertices if the corresponding chains have identical sequences. We analyze all possible maximum matchings in this graph using Uno's algorithm (1997), which enumerates perfect, maximum, and maximal matchings in bipartite graphs.

For the implementation, we used open-source code, which we refined by limiting it to the essential functions needed for our analysis. The source implementation is available at https://github.com/Xunius/bipartite_matching.

This approach enhances our tool's ability to handle complex assemblies with symmetric components, making it more versatile and applicable to a broader range of systems. Examples of RNA-RNA and protein-protein interactions are provided (in our Zenodo repository) to illustrate its application.

{Comment 5:} There is no support for mmCIF format, which is particularly relevant for this work which is supposed to work with large complexes with $>$65 chains.

{Response:} Thank you for your feedback on the support for the mmCIF format. We acknowledge that the lack of mmCIF format support is a limitation, especially for large complexes with more than 65 chains, as you mentioned. As noted in the manuscript, the current version of the I-INF tool operates on PDB files due to the limitations of the HBPLUS library. However, we have already planned to implement support for the mmCIF format in future updates to address this limitation and improve the tool's applicability for larger RNA-protein complexes. We appreciate your valuable suggestion, and we will continue to work toward improving the capabilities of the tool in future versions.

Reviewer 2

This work studies the evaluation of 3D structural prediction for macromolecular complexes, which is an essential problem in understanding cellular functions. The authors propose a novel metric, I-INF (Intermolecular Interaction Network Fidelity), to quantitatively assess the intermolecular interactions within multichain complexes. This manuscript is very well written and the I-INF score is useful. And I have only one minor comment about it. When comparing I-INF with existing methods such as TM-score and DockQv2, it is recommended to add a figure to make the comparison more clear.

{Response:} We thank the Reviewer for the positive feedback and helpful suggestions. In response, we have added two scatter plots (Figure 3) to the manuscript that visualize the correlations between TM-score, DockQv2, and I-INF. Additionally, we have included a violin plot in the Supplementary Information (S1\_Figure.pdf) to illustrate how the values of these measures are spread across the benchmark set. Each violin corresponds to one measure, highlighting its central tendency and variability. We believe these additions enhance the clarity of our evaluation and strengthen the presentation of our findings.

---

## [Decision Letter · Decision Letter 1]

7 Feb 2025

PONE-D-24-53421R1Assessing interface accuracy in macromolecular complexesPLOS ONE

Dear Dr. Szachniuk,

Thank you for submitting your manuscript to PLOS ONE. After careful consideration, we feel that it has merit but does not fully meet PLOS ONE’s publication criteria as it currently stands. Therefore, we invite you to submit a revised version of the manuscript that addresses the points raised during the review process.

Thank you for your revised manuscript and your response to the reviewer’s comments. However, the reviewer has pointed out that while their concern regarding the correlation between TM-score and I-INF was addressed in the response letter, no corresponding modifications were made in the manuscript itself.

You should incorporate a theoretical justification within the manuscript explaining why TM-score has a high correlation with I-INF. This addition should ensure that the manuscript provides sufficient context and rationale for the use of these metrics in evaluating complex assembly.

Please revise the manuscript accordingly and highlight the changes in your response letter.

We look forward to receiving your revised manuscript.

Kind regards,

Yong Wang

Academic Editor

PLOS ONE

Journal Requirements:

Reviewers' comments:

Reviewer's Responses to Questions

**Comments to the Author**

1. If the authors have adequately addressed your comments raised in a previous round of review and you feel that this manuscript is now acceptable for publication, you may indicate that here to bypass the “Comments to the Author” section, enter your conflict of interest statement in the “Confidential to Editor” section, and submit your "Accept" recommendation.

Reviewer #1: (No Response)

2. Is the manuscript technically sound, and do the data support the conclusions?

Reviewer #1: Partly

3. Has the statistical analysis been performed appropriately and rigorously? 

Reviewer #1: Yes

4. Have the authors made all data underlying the findings in their manuscript fully available?

Reviewer #1: Yes

5. Is the manuscript presented in an intelligible fashion and written in standard English?

Reviewer #1: Yes

6. Review Comments to the Author

Reviewer #1: I previously commented that "TM-score an I-INF does NOT measure the same aspect of complex assembly. I-INF only evaluate the correctness of interface, while TM-score evaluates the global topology. There should be some theoretical justification on why TM-score has a high correlation with I-INF." This revision only answer my question in the response letter, but did not make any corresponding modification in the manuscript.

7. PLOS authors have the option to publish the peer review history of their article (what does this mean?). If published, this will include your full peer review and any attached files.

Reviewer #1: No

---

## [Author Response · Author response to Decision Letter 2]

7 Feb 2025

We thank the Reviewer for the supplementary suggestion. We have answered it in the revised version of the manuscript, and below we provide detailed explanations:

Reviewer's comment:

I previously commented that 'TM-score and I-INF do NOT measure the same aspect of complex assembly. I-INF only evaluates the correctness of the interface, while TM-score evaluates the global topology. There should be some theoretical justification on why TM-score has a high correlation with I-INF.' This revision only answered my question in the response letter, but did not make any corresponding modification in the manuscript.

Response:

Indeed, in the previous version of the manuscript, we did not include the appropriate modification in response to this comment. We have now addressed this. In the "Results and Discussion" section, we have inserted the following text:

“Although TM-score evaluates the global topology of the model and I-INF specifically assesses the accuracy of the intermolecular interface, we observed a high Pearson correlation (0.73) between these metrics. This suggests that, in our dataset, a correctly predicted global fold is largely a consequence of the proper spatial arrangement of the molecular components, resulting in accurately modeled interfaces. In other words, deviations in the overall fold are primarily associated with errors in the interface regions. Therefore, high TM-score values frequently coincide with high I-INF values, highlighting our models' interdependence between global structural accuracy and interface correctness.”

We hope that this addition provides the necessary theoretical justification for the observed high correlation between TM-score and I-INF.

---

## [Editor Report · Decision Letter 2]

11 Feb 2025

Assessing interface accuracy in macromolecular complexes

PONE-D-24-53421R2

Dear Dr. Szachniuk,

We’re pleased to inform you that your manuscript has been judged scientifically suitable for publication and will be formally accepted for publication once it meets all outstanding technical requirements.

Kind regards,

Yong Wang

Academic Editor

PLOS ONE

---

## [Editor Report · Acceptance letter]

PONE-D-24-53421R2

PLOS ONE

Dear Dr. Szachniuk,

I'm pleased to inform you that your manuscript has been deemed suitable for publication in PLOS ONE. Congratulations! Your manuscript is now being handed over to our production team.

Kind regards,

on behalf of

Dr. Yong Wang

Academic Editor

PLOS ONE